# Thermodynamic and Spectroscopic Studies of SDS in Cinnamaldehyde + Ethanol Mixtures: Influences of Temperature and Composition

Waleed M. Alamier [1], Shadma Tasneem [1,*], Arshid Nabi [2], Nazim Hasan [1] and Firdosa Nabi [2]

1   Department of Chemistry, Faculty of Science, Jazan University, Jazan P.O. Box 114, Saudi Arabia
2   Department of Chemistry, Faculty of Science, University of Malaya, Kuala Lumpur 50603, Malaysia
*   Correspondence: sthaque@jazanu.edu.sa

**Abstract:** The study of intermolecular interactions between ethanol (E-OH), cinnamaldehyde (CAD) with anionic surfactant sodium dodecyl sulfate (SDS) in non-aqueous media has been examined by utilizing conductometric and spectroscopic techniques. The critical micelle concentration (CMC) values have been determined. The experimental conductance data were analyzed against temperature and concentration using standard relations. The pseudo phase separation model has been adopted to calculate various thermodynamic parameters like standard free energy, $\Delta G^{\circ}{}_{mic}$, enthalpy, $\Delta H^{\circ}{}_{mic}$, and entropy, $\Delta S^{\circ}{}_{mic}$, of micelle formation. Fourier transforms infrared analysis (FTIR), and Fluorescence spectra were taken out to assess the possible interactions prevailing in the micellar systems. The findings demonstrated that the presence of SDS, and the composition of CAD + ethanol might affect the thermodynamic parameters. The discrepancy in these parameters with the surfactant concentration or with the temperature change indicates the manifestation of different interactions prevailing in the studied systems.

**Keywords:** ethanol; CAD; thermodynamic parameters; CMC; hydrophobic–hydrophobic interactions; FTIR analysis

## 1. Introduction

The studies of interactions between surfactants and bioactive molecules are of enormous importance because of their applications in biological systems, pharmaceuticals, and biotechnological processes [1,2]. Surfactants are fascinating compounds because of their amphiphilic nature, hence they possess both hydrophilic and hydrophobic characteristics in the same mixtures and have soluble properties in polar and non-polar solvents [3]. Surfactants have the ability to form aggregates at a particular concentration in both aqueous and non-aqueous media. The concentration, at which the aggregation originates, is known as critical micelle concentration [CMC], and the aggregate is called a micelle. The phenomenon of aggregation in molecules incorporates both attractive and repulsive interactions. The nature of hydrophilic and hydrophobic moieties determines how bio-active compounds interact during formulation, agrregation and biological point of view in drug –surfacatnt interactions. Because of the presence of substituents on the hydrophobic core or variations of hydrocarbon chain length, it is evident that in solution systems, the behavior of such molecules can be varied [4].

To understand the intermolecular interactions of SDS in CAD + E-OH mixtures, we determined several physicochemical characteristics. The prevailing molecular interactions and micelle growth at particular concentrations are interpreted using different techniques such as conductivity, surface tension, refractive index, Uv-Visible spectroscopy and fluorescence [5]. However, we use conductivity and spectroscopic methods in this study; SDS was chosen because it is widely known for its ability to create micellar solutions. SDS is amphiphilic and features a 12-carbon tail connected to the sulfate group. It is said to be

more effective than urea and guanidine hydrochloride at denaturing proteins. Because of its high foaming properties, ease of accessibility, and low production cost, SDS is considered the best option. SDS has numerous biological functions, including antibacterial and skin cleaning effects [6]. It is well-documented that the interaction due to the presence of an oxygen group acts as both an acceptor and a donator [7]. In this work, SDS is chosen to quantify the impact of various solution environments on physical characteristics in the continued development of innovative drug delivery systems, including micellar solution.

On the other hand, cinnamaldehyde [CAD] is a liquid aldehyde derived from the bark of Cinnamomum trees, a naturally occurring flavonoid often used as a natural flavoring and fragrance agent in the kitchen and industry [8]. In addition, CAD is believed to have various beneficial properties, for instance antimicrobial [9], antioxidative [10], and inhibiting antiapoptotic [11] properties. CAD is found to protect against gram-positive/negative infection [1], diabetes [12], gastric ulcer [13], cardiomyopathy hypertrophy, [14] brain I/R injuries [15]. Cinnamaldehyde (CAD) is a representative substrate containing the carbonyl and vinyl groups simultaneously. It is worth mentioning, that the proposed work was designed to increase the solubility of hydrophobic CAD and to encapsulate the CAD molecules via micelles to offer protection against degradation. However, here we study the possible molecular interaction and micelle formation between CAD molecules in SDS ethanol mixed media. This study has further applications in drug delivery vehicles as an encapsulated micelle formulation, and is expected to improve cellular uptake with reduced side effects. However, the slight solubility of CAD in water restricts the study of interactions in aqueous media. Meanwhile, ethanol, a polar solvent self-associated through hydrogen bonding, is expected to interact strongly with other fluids by hydrogen-bonding, and was preferred in the present study. Thus, it would be interesting to make a comparison of the micellar properties of SDS in pure ethanol mixtures and in ethanol–CAD mixtures.

CAD is usually used as a flavoring agent, as well as a fragrance agent in global market like cosmetics, soaps, detergents, deodorants, shampoos, etc. [10–15]. Thus, it is essential to develop a simple, sensitive and selective analytical approach to detect the interaction properties of CAD. So, overwhelmed with its multiple applications in the diverse field, this work proposed a simple technique to interpret possible molecular interactions of CAD with surfactant in ethanol media using critical micelle concentration of SDS and spectroscopic techniques. Conductivity and thermodynamic empirical equations at different temperatures were used for molecular interaction parameters analysis.

In the present study, we evaluate the micelle formation of sodium dodecyl sulfate (SDS) with CAD molecules in ethanol media by conductometric, and spectroscopic methods. It is worth observing the effect of aggregation in bioactive molecules. Interestingly both SDS and CAD have different polarity regions with hydrophilic and hydrophobic substitutional groups of corresponding characteristics. Thus, it is meaningful to inspect the process of surfactant aggregation in a non-aqueous media.

## 2. Experiment

### 2.1. Materials

Ethanol E-OH 99% (Sigma–Aldrich, St. Louis, MO, USA) was used without further purification. Cinnamaldehyde (CAD) with preeminent quality was purchased from Sigma Life Sciences (Sigma-Aldrich) and kept under low temperature before use. The sodium dodecyl sulfate (SDS) was purchased from Friedmann Schmidt chemicals, purity 95% as reported by the vendor. The chemicals were dried in vacuum over $P_2O_5$ for about 72 h at room temperature.

### 2.2. Methods

#### 2.2.1. Stock Solution Preparations

The stock solutions of CAD 0.05 m (mol kg$^{-1}$) in ethanol were prepared by stirring the solution mixture for about 8 h at room temperature. Samples of 0.001, 0.002, 0.003, 0.004, 0.005, 0.006, 0.007, 0.008, 0.009, 0.010, 0.011, 0.012, 0.013, 0.014, 0.015, 0.016, 0.017,

0.018 and 0.019 m SDS solutions were prepared for the experimental investigations. An electronic balance Shimadzu AY220, Japan, with precision of ±0.0001 g, was used for weighing the freshly prepared samples. The appropriate arrangements were made for storage for all concentrations to avoid evaporation. Moreover, the chemicals used were systematically described by their chemical structures: (a) sodium dodecyl sulfate (SDS), (b) Cinnamaldehyde (CAD), and (c) ethanol, shown in Figure 1.

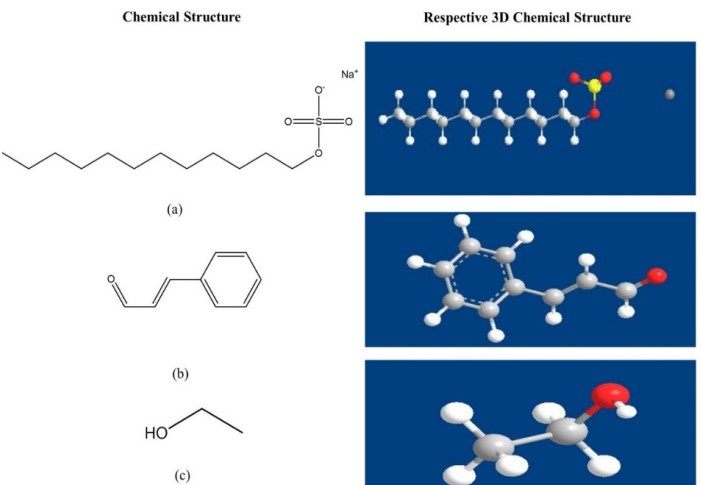

**Figure 1.** Chemicals and their respective 3D chemical structures of (**a**) sodium dodecyl sulfate, (**b**) cinnamaldehyde and (**c**) ethanol, respectively.

### 2.2.2. Conductivity Measurements

A digital conductivity meter (PC 510 Bench/Conductivity Meter, EUTECH Instruments) was used to determine the conductance of each sample. The conductivity meter was calibrated before measuring the conductance of the solutions using standard solutions of 0.01 and 0.10 N KCl (purity > 99%) prepared in doubly-distilled water, with a conductivity of 1413 µS/cm at 298.15 °K. The temperature of the water bath was maintained at a constant level for each measurement. To achieve thermal equilibrium and record conductivity data, the electrode assembly with glass tube was submerged in each sample solution. The conductivity measurements have a measurement error of 0.5%.

### 2.2.3. Fluorescence Measurement

To evaluate the interaction of cinnamaldehyde (CAD) and SDS, a fixed volume of ethanol solvent was separately mixed to dilute the solution. Each sample was incubated for five minutes at room temperature and then fluorescence spectra (Shimadzu RF-6000, Spectrofluorophotometer, Kyoto, Japan) were recorded at the excitation wavelength of 295 nm. For different detection range assessments, various sets of solutions of fixed volume were prepared by mixing ethanol. The fluorescence spectra were measured in a range from 200 nm to 400 nm after using an excitation wavelength of 295 nm. Every set of sample solution fluorescence emission spectra was recorded three times and evaluated. Cinnamaldehyde (CAD) quenched the maximum fluorescence intensity of SDS-ethanol that is further finely fitted in the Stern–Volmer equation, hence it is used in clinical science as an anticancer agent, stomachic, antipyretic and antiallergic drug [16]. The cinnamaldehyde (CAD) acts as a quencher to determine the sample mixture's binding constant ($K_b$). Furthermore, the interaction between the micellar solution of SDS and CAD with ethanol was also studied.

### 2.2.4. FTIR Measurement

A liquid-sample-based Fourier transform infrared (FTIR) spectrometer (Nicolet iS10, Thermo Fisher Scientific, Waltham, Mass, USA) was used to evaluate the interaction of CAD and SDS during pre- and post-micelle formation in an ethanol system. All measurements

were carried out using an attenuated total reflectance (ATR) accessory with a resolution of 4 cm$^{-1}$, and 15 scans of each sample were performed. A hardware specification of a flow-rate top plate fitted with a 421 ZnSe reflection crystal and a depth of penetration of 2 mm was used for spectral analysis.

### 2.3. Calculation of the Thermodynamic Properties of Micellization

The pseudo-phase separation model was used to interpret the thermodynamic parameters. As a result, the standard free energy of micellization can be calculated using the following Equations (1)–(6):

$$\Delta G_m^\circ = (2 - \beta)RT \ln X_{cmc} \tag{1}$$

where, $X_{cmc}$ is the *cmc* values expressed in mole fraction, $\beta$, is the degree of ionization of the micelles obtained by the ratio of two linear segments of conductivity (SDS + -E-OH) and (SDS + E-OH + CAD) plots above and below cmc values [16]. The other thermodynamic properties for instance enthalpy, $\Delta H_m^\circ$ and entropy, $\Delta S_m^\circ$ of micellization can be evaluated from the fundamental thermodynamic equations as:

$$\Delta H_m^\circ = RT^2(2 - \beta)\frac{d \ln X_{cmc}}{dT} \tag{2}$$

$$\Delta S_m^\circ = \frac{\Delta H_m^\circ - \Delta G_m^\circ}{T} \tag{3}$$

where, $\frac{d \ln X_{cmc}}{dT}$ values fitted to a polynomial function with the values of $d \ln X_{cmc} \Delta T$ as:

$$\ln X_{cmc} = a + b(T/K) + c(T/K)^2 \tag{4}$$

where, $a$, $b$ and $c$ are the polynomial constants, and the above equation can also be expressed as:

$$\frac{\ln X_{cmc}}{dT} = b + 2c(T/K) \tag{5}$$

Further, the results obtained from the above thermodynamic Equations (1)–(5) and the values presented in Table 1, provides the values of cmc, $\ln X_{cmc}$, $\beta$, $\Delta G_m^\circ$, $\Delta H_m^\circ$, $\Delta S_m^\circ$ and $T\Delta S_m^\circ$, respectively.

**Table 1.** The thermodynamic parameters of micellization at different temperatures (298.15 K to 318.15 K) for SDS + E-OH and SDS + E-OH + CAD, respectively.

| T (K) | CMC (M) | $\alpha$ | $\Delta G_m^\circ$ (kJ mol$^{-1}$) | $\Delta H_m^\circ$ (kJ mol$^{-1}$) | $\Delta S_m^\circ$ (J mol$^{-1}$ K$^{-1}$) | $T\Delta S_m^\circ$ (J mol$^{-1}$ K$^{-1}$) |
|---|---|---|---|---|---|---|
| | | | SDS + Ethanol | | | |
| 298.15 | 0.00822 | 0.65 | −29.51 | −13.06 | 0.055 | 16.9051 |
| 303.15 | 0.00881 | 0.64 | −29.88 | −13.56 | 0.054 | 16.3701 |
| 308.15 | 0.00939 | 0.60 | −31.16 | −14.47 | 0.054 | 16.6401 |
| 313.15 | 0.00995 | 0.59 | −31.68 | −15.05 | 0.053 | 16.5969 |
| 318.15 | 0.01073 | 0.56 | −32.50 | −15.83 | 0.052 | 16.5438 |
| | | | SDS + Ethanol + Cinnamaldehyde | | | |
| 298.15 | 0.0120 | 0.76 | −22.99 | −1.72 | 0.071 | 21.1686 |
| 303.15 | 0.0122 | 0.73 | −23.74 | −2.68 | 0.069 | 20.9173 |
| 308.15 | 0.0125 | 0.71 | −24.49 | −3.74 | 0.067 | 20.6460 |
| 313.15 | 0.0127 | 0.69 | −25.32 | −4.90 | 0.065 | 20.3547 |
| 318.15 | 0.0129 | 0.66 | −26.15 | −6.17 | 0.063 | 20.0434 |

## 3. Results and Discussion

### 3.1. Conductometric Study

Electrical conductivity techniques determine the interaction and association of molecules in diverse fluid systems [17,18]. In the current study, two sets of conductivity tests were performed to evaluate the interaction behavior and affiliation of different molecules in the component systems. The conductance was measured as a function of SDS in the presence of ethanol (Figure 2) and SDS in presence of CAD + ethanol in the concentration range of 0.001 to 0.020 mol kg$^{-1}$ at 298.15, 303.15, 308.15 and 313.15 K and are presented in Figure 3. The intersection of straight lines above and below the breaking points of conductance ($\kappa$) against surfactant concentration at different temperatures acquired the CMC values. The values of CMC for SDS in pure ethanol solutions are presented in Table 1. The data given in Table 1 presents the minimum value of CMC verses temperature curve for SDS + ethanol system at lower temperature (298.15 K). The CMC of SDS rises from 298.15 to 313.15 K as the temperature rises. The influence of temperature on CMC has been shown to be system dependent [19]. Depending on the type of solvent solution employed, the micellization process might either be instantaneous or sluggish [20]. In the instance of the SDS in CAD + ethanol solution, the specific conductivity values gave a distinct breakpoint suggesting instantaneous micellization.

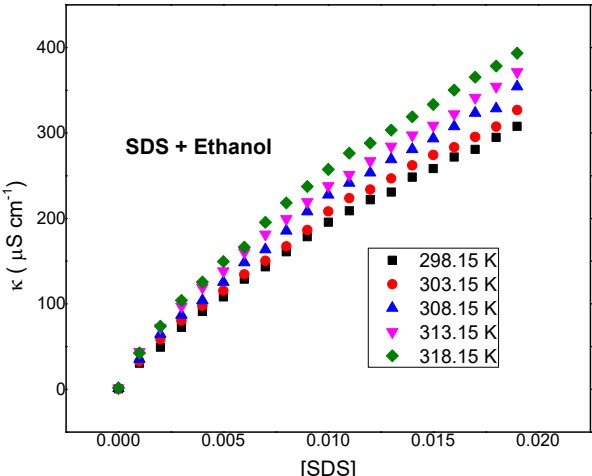

**Figure 2.** Specific conductance of SDS in ethanol at different temperatures.

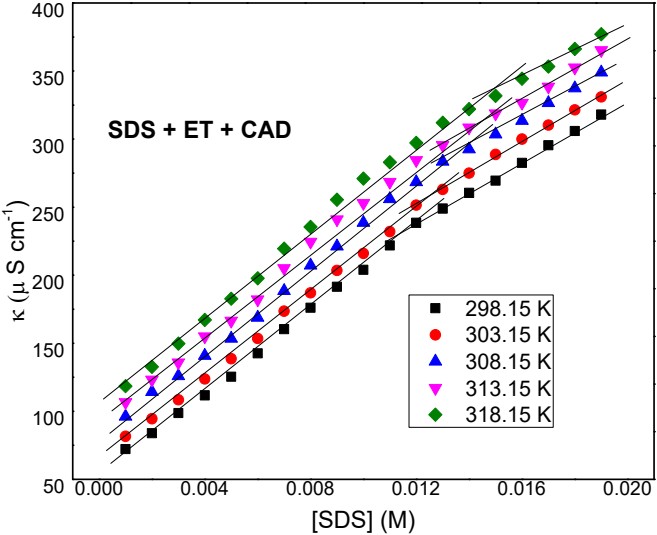

**Figure 3.** Specific conductance of SDS in cinnamaldehyde + ethanol at different temperatures.

However, In Figure 3, CMC values increase when CAD is introduced into the system, and the micellization process observed is different. CAD molecules are insoluble in an aqueous medium so we conducted the experiment using an ethanol medium. Ethanol acts as a structure-breaking solvent which causes the destruction of the inherent H-bond association of CAD and SDS. The incorporation of CAD molecules with SDS molecules resulted in slower micellization. This slight change in CMC values in the (SDS in CAD + Ethanol) system in Table 1 reveals that an addition of CAD molecules led to an effect on the types of interactions occurring in the system. However, micellization is delayed when CAD is added, which can be explained by the presence of an aromatic ring in the CAD's structure, which may establish a fine balance of hydrophobic interactions between SDS's long hydrocarbon tails and repulsive interactions between the ionic head groups. In the presence of CAD, a delay in CMC values was reported. The values of CMC rise as the temperature rises. Due to the existence of various components in the mixture, the effect of temperature on the magnitude of the CMC values of surfactant was frequently examined. The disruption of the ordered molecules surrounds the surfactant's hydrophobic groups at higher temperatures. Furthermore, it is advantageous that at higher temperatures, thermal motion increases, resulting in de-micellization due to the distraction of the micelle's palisade layer, which increases the CMC of the surfactants [21]. This is due to the fact that when the temperature rises, the high solubility of hydrocarbons stabilizes the surfactant monomers, preventing micelle formation, resulting in a larger SDS CMC [22].

The evaluations of the thermodynamic parameters of the pseudo-phase separation model were deduced from various thermodynamic micellization equations as described above. The Gibbs free energy of micellization, $\Delta G_m^\circ$, entropy of micellization, $\Delta S_m^\circ$ and enthalpy of micellization, $\Delta H_m^\circ$ are worth applying to understand molecular interaction and are also the core concepts behind the process of micellization [23]. Furthermore, the data obtained from the thermodynamic parameters are useful for determining the influence of environmental and structural contributions to CMC values, as well as observing the effects of novel structures and environmental deviations in the presence of various components.

Meanwhile, in an ionic surfactant it has been stated that the values of $\Delta G_m^\circ$ lie in the range from $-23$ to $-42$ kJ mol$^{-1}$ at 298.15 K [24]. It was noted that the values $-29.51$ kJ mol$^{-1}$ for SDS in ethanol and $-22.99$ kJ mol$^{-1}$ for SDS in cinnamaldehyde + ethanol at the corresponding temperature lie in the reported range. There were more negative values of free energy of micellization, $\Delta G_m^\circ$ in SDS with -E-OH mixtures than SDS + E-OH + CAD mixtures at all temperature from 298.15 to 313.15 K, observed experimentally. Moreover, the free energy of micellization $\Delta G_m^\circ$ examined in Table 1 is negative for both systems at all temperature from 298.15 to 313.15 K. Therefore, the values of $\Delta G_m^\circ$ of micellization are more for the SDS +E-OH system than in the SDS + E-OH + CAD system. This signifies the readiness and thermodynamically more spontaneous micellization of SDS with the E-OH mixture than the SDS + E-OH + CAD mixtures. It was noticed that the values of $\Delta G_m^\circ$ increase with an increase in temperature, suggesting that the process of micellizations in both studied systems was thermodynamically spontaneous. It was recognized that the desolvation of the hydrophilic groups of the surfactants took place if the $\Delta G_m^o$ values declined with increasing temperature [25].

Moreover, the observed increased negative values of $\Delta G_m^\circ$ with elevated temperatures are because of the mutual effects of the $\beta$ and $\ln X_{cmc}$ values. The $\beta$ values increased while the $\ln X_{cmc}$ values decreased at higher temperatures and vice versa. The reason for the increased values of $\beta$ with rising temperature is desolvation of the hydrogen bond which persisted among the H atoms of –OH groups of ethanol and the polar head groups of the SDS molecules. Sequentially, the surface charge density and electrostatic repulsions between head groups of SDS + E-OH + CAD molecules increased more with the elevated temperatures. Thus, it disfavors the process of micellization with increased temperatures. Similar to $\Delta G_m^\circ$ values, the values of $\Delta H_m^\circ$ are negative for all the systems at all temperatures. It is worth mentioning that the observed values of $\Delta G_m^\circ$, $\Delta H_m^\circ$ and $\Delta S_m^\circ$ in E-OH + SDS were found to be $-29.51$ kJ mol$^{-1}$, $-13.06$ kJ mol$^{-1}$ and $0.055$ kJ mol$^{-1}$ K$^{-1}$ at 298.15 K,

respectively, and for SDS + E-OH + CAD were found to be $-22.99$ kJ mol$^{-1}$, $-1.72$ kJ mol$^{-1}$ and 0.071 kJ mol$^{-1}$ K$^{-1}$, which are much lower than the literature values $-39.70$ kJ mol$^{-1}$, $-22.98$ kJ mol$^{-1}$ and 0.06 kJ mol$^{-1}$ K$^{-1}$ [26], respectively, of SDS in pure water at the corresponding temperatures. Lower values of SDS in ethanol than in pure water may be explained by considering the nature of the solvent (ethanol). The marked difference in the thermodynamic parameters may be because of the relative permittivity of water (e = 79.99 at 293.15 K) and ethanol (e = 25.02 at 293 K) [27] is expected to affect the thermodynamic parameters of SDS in these media. The values of standard free energy of micellization becomes less negative with the addition of CAD which indicates that the micellization is less favored with the addition of CAD in the SDS ethanolic media. Similar results have also been reported for the drug chloroquine with SDS in polar aqueous solution, where $\Delta G_m^{\circ}$, $\Delta H_m^{\circ}$ and $\Delta S_m^{\circ}$ values were found to be $-21.74$ kJ mol$^{-1}$, $-3.45$ kJ mol$^{-1}$ and 62.42 kJ mol$^{-1}$ K$^{-1}$ at 298.15 K, respectively [28]. The increased negative values of $\Delta H_m^{\circ}$ and the decreased positive values of $\Delta S_m^{\circ}$ (Table 1) weaken the hydrophobic interactions between the SDS and CAD molecules. In contrast, the electrostatic interactions become stronger with rising temperatures. The values of $\Delta H_m^{\circ}$ are believed to derive from the interactions of electrostatic, hydrophobic, or polar head group hydrations and because of the counter ion micelle bindings. Moreover, the investigated $T\Delta S_m^{\circ}$ values are much higher in magnitude as compared to the $\Delta H_m^{\circ}$ values. Hence, this indicates micellization is entropy driven, having the potential for hydrophobic groups of surfactants to orient themselves to the interior of micelle core from the bulk solvent [29]. Behind the possibility of micellization is the tendency of structure-breaking E-OH to act as a solvent on the surfactant (SDS) molecules. This may be possible because of the higher freedom of hydrophobic chains in the vicinity of the nonpolar interior of micelles than in the solvent environment [18,30].

### 3.2. Fluorescence Spectra Behaviour of Cinnamaldehyde (CAD) in SDS-Ethanol Media

Cinnamaldehyde (CAD + ethanol + SDS) exhibits the maximum fluorescence emission at 310 nm. The maximum emission was shown between 300–400 nm for SDS in ethanol + CAD. The study was further inferred by utilizing specified sets of parameters, such as to investigate the fluorescence intensities in a region before and after micellization [30]. The pre-micellar region of SDS with varying concentration in CAD + ethanol was inferred as depicted in Figure 4. In addition, it was observed that after a certain point i.e., 0.012 m concentration of SDS with CAD in ethanol, a linear decrease in fluorescence intensity with decreasing concentrations was attained. The observed results are believed to be due to the presence of more SDS monomers in the surfactant micellar system which further modifies monomer interactions in aggregates, thereby alter the fluorescence intensities in the premicellar and postmicellar region. In premicellar region, CAD molecules possibly interact onto the surface of SDS via hydrogen bonding between the head group of SDS and substitute (H-C=O) groups of CAD molecules resulting in aggregates rather than micelles. In other words, with an increase in the concentration of SDS molecules to CAD in an ethanol solution, more aggregates are possibly formed leading to increase in fluorescence intensities up to a concentration i.e., critical micellar concentrations (CMC). In addition, at the micellar region a sudden jump was observed in fluorescence intensity, leading to a constant decrease in fluorescence intensities, with an increase in the concentration of SDS with CAD in ethanol media. However, in the post-micellar concentration another rearrangement occurs between the SDS monomers and drug molecules in ethanol media. The maximum fluorescence intensity was observed at 0.019 m concentration in the post-micellar region as depicted in Figure 5. Micellar systems consisting of a surfactant and an additive such as an organic compound usually self-organize as a series of worm-like micelles that ultimately form a micellar network. The nature of the additive influences micellar structure and properties such as aggregate lifetime. CAD molecules show $\pi$–$\pi$ stacking interaction that somewhat segregates into SDS micelle at higher micellar concentrations [29,30]. It is believed that at higher micellar concentrations, the hydrogen bonding forces onto the surface between CAD and SDS monomers become more favorable and

organize as micelle assemblies in comparison to the sole structural forces of the solvent i.e., ethanol molecules. In addition, the hydrophilic counterpart of the CAD molecules does not intervene in the micellar growth. Therefore, the fluorescence quenching of SDS with the cinnamaldehyde is mainly achieved by photo-excited electron transfer between the micelles of SDS and cinnamaldehyde molecules, and the process is thermodynamically feasible with a $\Delta G$ value of $-4.30 \times 10^4$ kcal mol$^{-1}$. The maximum fluorescence intensity of SDS is quenched gradually with the cinnamaldehyde in ethanol and the quenching efficiency of CAD can be finely fitted with the Stern–Volmer equation. Fluorescence measurement offers the information about the binding process during the self-aggregation process of SDS in the CAD-ethanol system. The fluorescence spectra of the observed pre- and post-micellar regions are shown in Figures 4 and 5 at maximum fluorescence excitation ($\lambda_{ex}$ 295 nm). The fluorescence intensities increase in the pre-micellar region whereas with the increase of SDS concentration the intensities decrease steeply resulting in the fluorescence quenching. Quenching can appear as a result of micellization depicting the role of hydrophobic interaction in the binding of molecules with the micelles. Fluorescence quenching is the decrease of the quantum yield of fluorescence from a fluorophore induced by a variety of molecular interactions with quencher molecules [31]. Therefore, the fluorescence data below the CMC and above the CMC are well fitted in the Stern–Volmer equation as given below:

$$\frac{F^\circ}{F} = 1 + K_q \tau^\circ \, [Q] = 1 + K_{SV} \, [Q], \tag{6}$$

where, F$^\circ$ and F are the fluorescence intensities in the absence and presence of quencher, respectively. A linear regression plot is drawn between the F$^\circ$/F against the molarity of varying concentration of the quencher [Q] (Figures 6 and 7). $K_{sv}$ is the Stern–Volmer constant, and its values are listed in Table 2. Below the CMC, the plot is linear, showing lower value of $K_{sv}$ than in the post-micellar region, i.e., above the CMC, suggesting that the micellization is controlled by the dynamic quenching mechanism rather than the static mechanism. $K_q$ is the quenching rate constant, and the values are evaluated by using Equation (6) [32] in the following form:

$$K_q = \frac{K_{SV}}{\tau^\circ} \tag{7}$$

where, $\tau^\circ$ is the average lifetime of the fluorescent molecule with the value ($10^{-8}$ s$^{-1}$), and the $Kq$ values estimated from Equation (7) are presented in Table 2. Hydrophobicity of the SDS molecule in the pre- and post-micelle regions has a significant influence on the binding constant, and the equilibrium between free and bound molecules is given by the Equation (8) [32].

$$\mathrm{Log}_{10} \frac{F^\circ - F}{F} = \log_{10} K_b + n \, \log_{10} \, [Q] \tag{8}$$

where, $K_b$ is the binding constant and $n$ is the binding sites, and the value of $K_b$ and $n$ is obtained from the intercept and slope of the linear regression plot between $\log_{10}$ ((F$^\circ$ − F)/F) versus $\log_{10}$ [Q] below and above the CMC. The values are shown in Table 2. The higher value of $K_b$ above the CMC than the binding constant below the CMC is positive evidence of the hydrophobic interaction. The observed comparatively lower fluorescence intensities in pre-micellar concentrations occurred after the repulsive electrostatic forces and dipolar interactions between CAD and SDS molecules in the ethanol medium, in turn resulting in a reduction of the fluorescence intensities. In addition, at pre-micellar concentrations, the partitioning of drug molecules has maximum probabilities as more forces are prevailing to form fewer aggregates in comparison with the micellar region hence, hydrophobic forces are predominating major driving force for micellization in the micellar region.

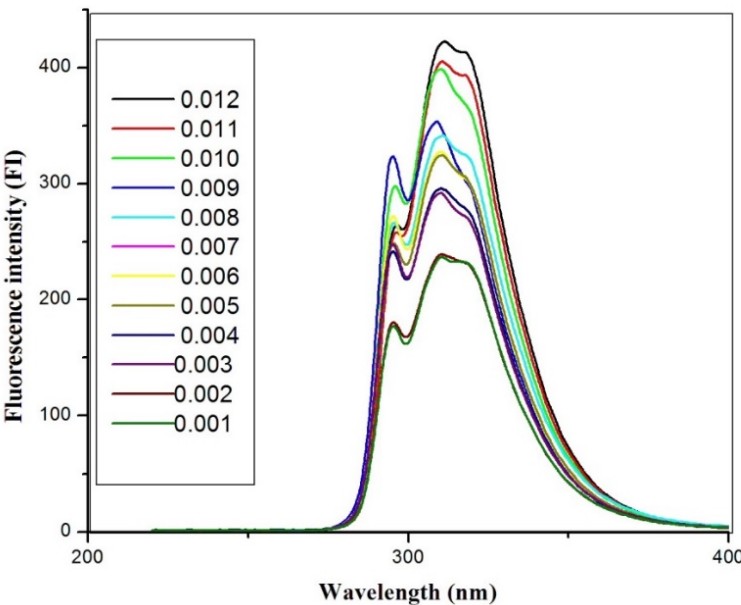

**Figure 4.** Fluorescence intensities (FI) versus wavelength for pre-micellar concentrations of SDS in CAD + E-OH.

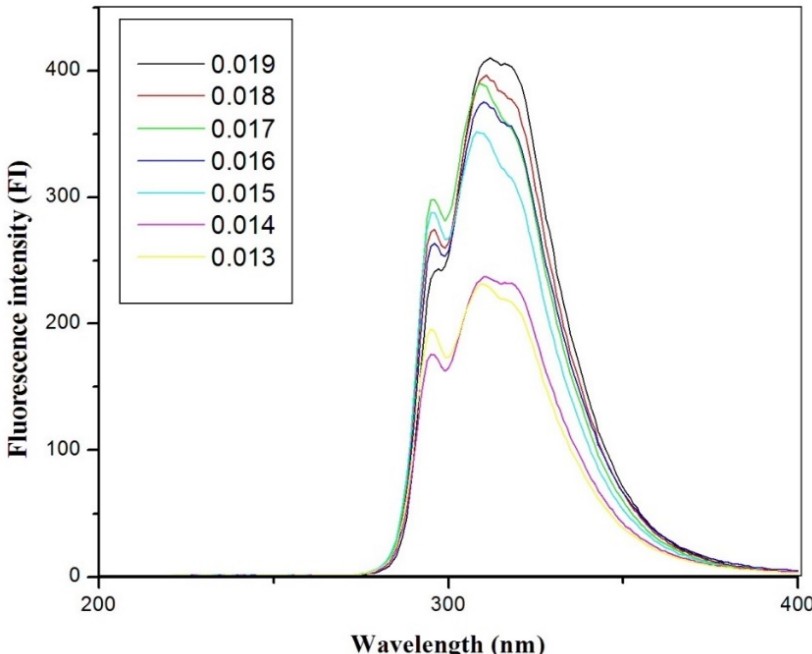

**Figure 5.** Fluorescence intensities (FI) versus wavelength for post-micellar concentrations of SDS in CAD + E-OH.

By knowing the binding constant of SDS with the CAD-ethanol system below and above the CMC, thermodynamic parameter free energy change ($\Delta G°$) can also be evaluated easily in the surfactant transition process of monomer to micellar phase by the following equation [32]:

$$\Delta G° = -RT \ln K_b \tag{9}$$

The obtained negative values of free energy change ($\Delta G°$) (listed in Table 2) signify the SDS thermodynamic stability and governs the binding process favoring the micellization process.

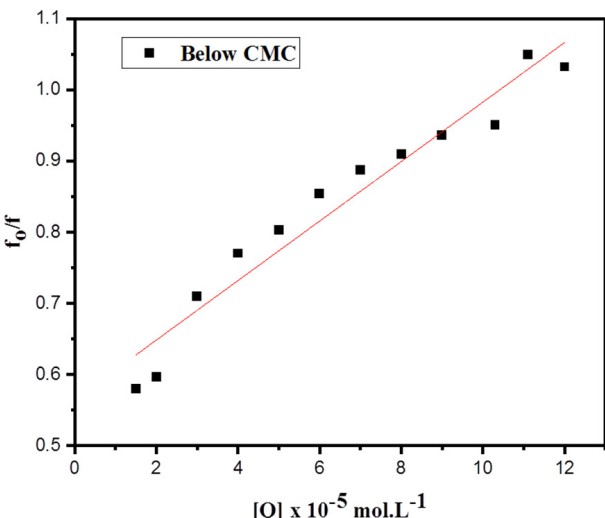

**Figure 6.** Stern–Volmer plot of interaction of SDS in the CAD-ethanol system at 298.15 K below CMC.

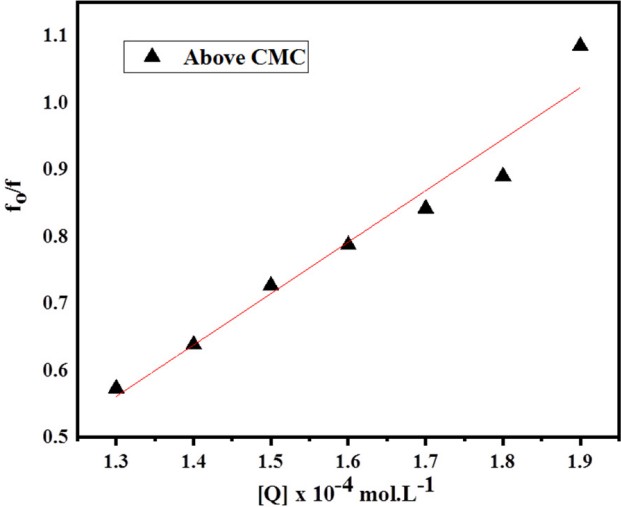

**Figure 7.** Stern–Volmer plot of interaction of SDS in CAD-ethanol system at 298.15 K above CMC.

**Table 2.** Stern–Volmer quenching constants ($K_{sv}$), quenching rate constant ($K_q$), binding constants ($K_b$), number of binding sites ($n$), and Gibbs free energy ($\Delta G°$) for the interaction of cinnamaldehyde (CAD) in SDS-ethanol media at 298.15 K temperature.

| (Measured at $\lambda_{ex}$ 295 nm) | Below CMC | Above CMC |
|---|---|---|
| $K_{sv}$ ($10^3$ M$^{-1}$) | 4.19 | 7.68 |
| $K_q$ | $4.5 \times 10^{10}$ | $3.3 \times 10^{15}$ |
| $K_b$ (M$^{-1}$) | $4.51 \times 10^2$ | $3.34 \times 10^7$ |
| $n$ | 1.22 | 4.38 |
| $\Delta G°$ ($10^4$ kJ mol$^{-1}$) | −1.51 | −4.30 |

### 3.3. FTIR Analysis of CAD + SDS + E-OH Media

The FTIR analysis was inferred to interpret the possible interactions in an ethanol medium between the cinnamaldehyde (CAD) and sodium dodecyl sulphate (SDS) molecules. The presence of different substitutions with their available surrounding environment between different molecules influences molecular interactions in their aqueous media [33]. Hereby, we determined the possible existing molecular interactions between the CAD

and SDS as deduced from both the structural information of pure CAD, pure SDS, and varying concentrations of CAD + SDS in an ethanol medium as depicted in Figure 8. Some of the observed essential peak intensities were lower, which confirms the existence of major functional groups of pure cinnamaldehyde. The intensities at 3062.6 cm$^{-1}$ and 3026.7 cm$^{-1}$ are attributed to presence of -C-H structuring vibrations and aromatic C–H bond structuring vibrations, 2814.5 cm$^{-1}$ and 2742.6 cm$^{-1}$ are because of –CH$_2$ structuring vibrations, and the presence of peak intensities within the range of 1648 cm$^{-1}$ to 1746 cm$^{-1}$, such as 1713.0 cm$^{-1}$, 1667.8 cm$^{-1}$, and 1619.7 cm$^{-1}$ are corresponding -C=O vibrations; the presence of peak intensities between 1463 cm$^{-1}$ and 1627 cm$^{-1}$, such as 1495.7 cm$^{-1}$ and 1574.5 cm$^{-1}$, are attributed to aromatic ring –C=C structural structuring vibrations. Meanwhile, C=C of aromatic ring vibrations were observed at 687.5 cm$^{-1}$; the presence of these peak intensities is attributed to pure cinnamaldehyde (Figure 6 (b; black line)) [34]. In addition, the major peak intensities as observed at 3324.6 cm$^{-1}$, 2971.6 cm$^{-1}$, 2926.4 cm$^{-1}$, 2881.2 cm$^{-1}$, 1085.8 cm$^{-1}$, and 1015.2 cm$^{1}$, respectively were observed in the pure spectrum of SDS molecules as depicted in Figure 6 (c; red line). Observed peak intensity at 3324.6 cm$^{-1}$ was assigned to the -O-H structuring vibrations modes, 2971.6, 2926.4, and 2881.2 cm$^{-1}$, which are attributed to the presence of asymmetric and symmetric structuring vibration modes of C-H aliphatic carbon chains [35]. In addition, the observed peak intensities at 1085.8 and 1015.2 cm$^{-1}$ are assigned to the corresponding S-O and S=O structuring vibrational modes respectively [36,37]. Based on the pre- and post-micellization concentrations of the FTIR analysis of combined CAD + SDS in ethanol solution as depicted in Figure 6 (F; cyan line and G; pink line), we observed the disappearance of CAD peak intensity at 1667.6 cm$^{-1}$ (i.e., presence of -C=O vibrations), and also the observed lengthening of -O-H structuring vibrations at 3324.6 cm$^{-1}$ towards the higher frequency, i.e., 3327.0 cm$^{-1}$ in mixed CAD + SDS, emphasizing the possible hydrogen bond interaction in the mixed ethanol medium. The observed results of the FTIR analysis further established the aggregation of CAD with SDS molecules in the pre-micellar region and the formation of micelle assemblies. Additionally, in the pre- and post-micellar regions the possible hydrophobic and hydrogen bonding interaction between CAD and SDS are established after their function group interactions in an ethanol environment.

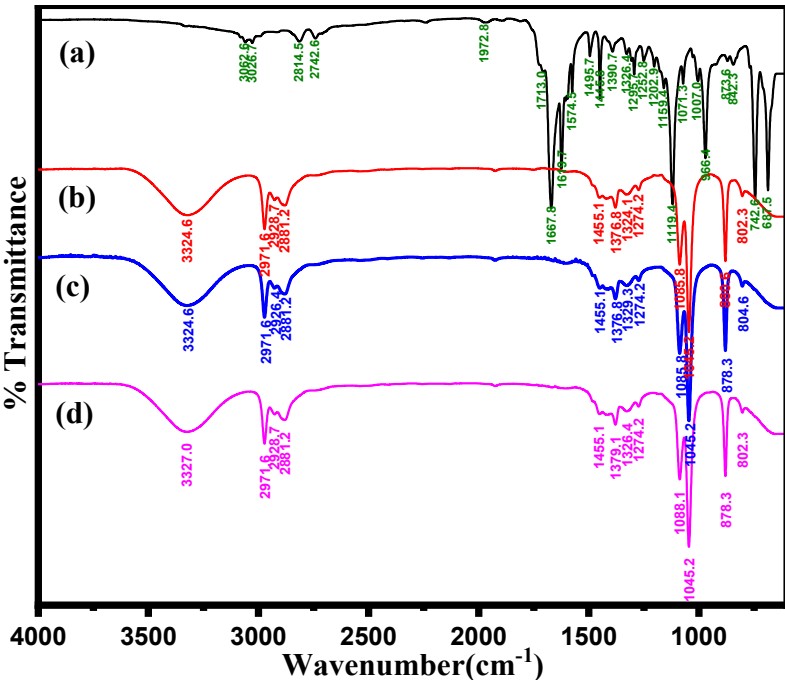

**Figure 8.** FTIR analysis of CAD + SDS at varying concentrations: (**a**) FTIR analysis of CAD, (**b**) SDS alone in ethanol, (**c**) Pre–micellization SDS + E-OH + CAD (**d**) Post-micellization SDS + E-OH + CAD.

## 4. Conclusions

The present study was conducted to deduce the possible molecular interaction between SDS and CAD molecules in an ethanol medium. The result obtained from the conductivity measurements implies delayed micellization after the molecular interactions in SDS + E-OH + CAD. The observations established the fine balancing hydrophobic force between the aromatic ring of CAD molecules and SDS's long hydrocarbon tails. whereas repulsive interactions also encounter the ionic head groups. Meanwhile, in SDS + E-OH and SDS + E-OH + CAD with elevated temperatures, increased thermal motions are quite possible with the distraction of the micelle's palisade layer, and in turn results in de-micellization, as observed with the increased CMC values of surfactant molecules. The values of $\Delta G_m^\circ$ of micellization were more for the SDS + E-OH system than in the SDS + E-OH + CAD system. This signifies the readiness and thermodynamically more spontaneous micellization of SDS with E-OH mixture than SDS + E-OH + CAD mixtures. In addition, increased numerical negative values of $\Delta H_m^\circ$ and the decreased numerical positive values of $\Delta S_m^\circ$ prompt the hydrophobic interactions between the SDS and CAD molecules weaken, while the electrostatic interactions become stronger with rising temperatures.

Moreover, from fluorescence analysis we believe the hydrogen bonding forces onto the surface between CAD and SDS monomers become more favorable and organize as micelle assemblies in comparison to the sole structural forces of the solvent i.e., ethanol molecules. Furthermore, lower fluorescence intensities were observed at lower micellar concentrations after the repulsive electrostatic forces and dipolar interactions between CAD and SDS molecules in the ethanol medium. Our observation of the molecular interaction and micelle growth was further supported upon an FTIR functional group analysis, from different peak intensities at varying concentration, in the pre- and post-micellar region. The possible hydrophobic and hydrogen bonding interactions between CAD and SDS were established as a regard of consequence of function group interactions in an ethanol medium.

**Author Contributions:** Conceptualization, W.M.A. and N.H.; methodology, W.M.A. and N.H.; software, S.T.; validation, S.T. and N.H.; formal analysis, S.T. and N.H.; investigation, S.T.; resources, W.M.A.; data curation, S.T.; writing—original draft preparation, A.N.; writing—review and editing, A.N., S.T. and F.N.; visualization, S.T.; supervision, S.T. and W.M.A.; project administration, N.H.; funding acquisition, W.M.A. All authors have read and agreed to the published version of the manuscript.

**Funding:** This research was funded by Deanship of Scientific Research, Jazan University (Grant No. RUP2-02).

**Institutional Review Board Statement:** Not applicable.

**Informed Consent Statement:** Not applicable.

**Acknowledgments:** The authors extend their appreciation to Deanship of Scientific Research Work, Jazan University, for supporting this research through the Research Units Support Program, Support Number: RUP2-02.

**Conflicts of Interest:** The authors declare no conflict of interest.

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
