# Peer review of "Thermodynamic and Spectroscopic Studies of SDS in Cinnamaldehyde + Ethanol Mixtures: Influences of Temperature and Composition"

_applsci, doi:10.3390/app122312020_

Round 1

Reviewer 1 Report

Authors have done a simple but very good work. It has a great important value in this field. The manuscript is well written.

Special Comments:

Author should measure/determine the binding constant theoretically (if possible).

Units of energy should represent in proper way.

Authors must cite followings articles in these regards-

1.      Chemical Physics Letters 694 (2018) 7–13

Author Response

NA

Reviewer 2 Report

Dear Authors

Where is the novelty?

you stated that in the study you "assess the micellization of sodium dodecyl sulfate (SDS) with ethanol in presence of CAD by conductometric, and spectroscopic methods." I am sorry, but you did not do it, which is why I cannot recommend this work for publication.

shifting of 3/cm on OH stretching region cannot be analyzed as hydrogen-bond formation. More in Fig. 3 there is 3327.0, not 3327.6 1/cm; so someone forgot to correct the number in the spectrum?

IR, as well as fluorescence analysis, prove nothing,

more, I think via mistake the fluorescence data you have shown are from other studies.

Additionally, the methodology section must be highly corrected, it is not understandable

Finally, what was the goal of this work?

Author Response

NA

Reviewer 3 Report

This manuscript deals with the investigation of intermolecular interactions between ethanol (E-OH), cinnamaldehyde (CAD) with sodium dodecyl sulfate (SDS) in non-aqueous media by using conductometric and spectroscopic techniques. Based on the importance of the topic the manuscript is acceptable for the publication after the MAJOR REVISION of the following:

1.       English grammar needs to be checked

2.       The authors need to provide more information regarding the grade of the ethanol used. It is known that 99% ethanol technical grade contains the traces of benzene and, thus, this trace benzene may be involved in the micelle formation process/interaction with CAD. There some research dealing with determination of benzene in ethyl alcohol Anal. Chem. 1948, 20, 11, 1063–1065). I may also suggest to carry out some blank experiments in food grade ethanol  containing 1 -  5% of water to see the influence of water on micelle formation process.

Author Response

NA

Round 2

Reviewer 2 Report

Dear Authors, 4 remarks and nothing:

1.     I asked about novelty: in the answer you characterized CAD; in SCOPUS “TITLE-ABS-KEY (cinnamaldehyde)” 7681 results, Among them, there are a lot of reports including changes in solubility, hydrophobicity, and pharmacologic response; so again where is the novelty?

2.     In the text p.13 is 3327.6/cm while in Fig 8 – 3327.0/cm, only this - no more

3.     Proofs: the scientific text needs support in the Figures; even the best text without such props, is only wishful thinking; by showing the whole IR range of the spectrum or bare FS spectra (only) no one can see your proofs.
More, much more important than the name of Apparatus is the technique used, by using IS10 you most probably work in ATR (what about the crystal used? What about resolution?)
How can you be sure that the signals ca. 3330/cm are from NH and not OH from EtOH????
You tried to correct somehow your FS results, but you have to show besides the bare FS the quenching effect,
More, from Figs 4-5, and SM4-5, one can read that your excitation wavelength is ca. 290 nm, not 310 nm; while the emission (NOT - maximum emission!!! Because this is ca. 320 -330 nm) was shown between 300 – 400 nm (NOT 300 – 450 nm)
(is it now more
„scientific language”?)

4.     Experimental section needs much more corrections (see also above)
- what do you mean: “
Small samples were taken in demountable cells, and three times spectral scan was recorded.”? 3-spectra of the same sample? Why?
- p5 you wrote: “
were recorded at the excitation wavelength of 270 nm
- p10 & p11 & in Tab.2  - “Measured at λex 310 nm” – the fact is, that in Figs 4 & 5 the excitation is 290 nm!!!!
- there is something wrong, can you please show the pure-SDS fluorescence spectrum?
- Figs. 4-7 the numbers mean (I guess) concentration, of what? I've lost: CAD? SDS? What is a quencher of what?

Author Response

Thank you.

Reviewer 3 Report

The authors have improved their manuscript according to the suggestions and fully addressed my comments. The manuscrits is Acceptable in its present form.

Author Response

Thank you sir to accept our manuscript.

Round 3

Reviewer 2 Report

---